# Cafestol and Kahweol: A Review on Their Bioactivities and Pharmacological Properties

**DOI:** 10.3390/ijms20174238

**Published:** 2019-08-30

**Authors:** Yaqi Ren, Chunlan Wang, Jiakun Xu, Shuaiyu Wang

**Affiliations:** 1College of Veterinary Medicine, China Agricultural University, Beijing 100193, China; 2Key Laboratory of Sustainable Development of Polar Fisheries, Ministry of Agriculture and Rural Affairs, Yellow Sea Fisheries Research Institute, Chinese Academy of Fishery Sciences; Laboratory for Marine Drugs and Byproducts of Pilot National Laboratory for Marine Science and Technology (Qingdao), Qingdao 266071, China

**Keywords:** cafestol, kahweol, serum lipid, anti-inflammation, anti-tumor, anti-diabetes

## Abstract

Cafestol and kahweol are natural diterpenes extracted from coffee beans. In addition to the effect of raising serum lipid, in vitro and in vivo experimental results have revealed that the two diterpenes demonstrate multiple potential pharmacological actions such as anti-inflammation, hepatoprotective, anti-cancer, anti-diabetic, and anti-osteoclastogenesis activities. The most relevant mechanisms involved are down-regulating inflammation mediators, increasing glutathione (GSH), inducing apoptosis of tumor cells and anti-angiogenesis. Cafestol and kahweol show similar biological activities but not exactly the same, which might due to the presence of one conjugated double bond on the furan ring of the latter. This review aims to summarize the pharmacological properties and the underlying mechanisms of cafestol-type diterpenoids, which show their potential as functional food and multi-target alternative medicine.

## 1. Introduction

Coffee is one of the most wildly consumed beverages in the world. Extensive experiments and meta-analysis of studies show that coffee consumption is associated with the risk of some human diseases [1]. Statistics show that moderate consumption of coffee (3 to 5 cups/day) reduces the risk of cardiovascular disease [2]. There is growing evidence that coffee can prevent some chronic diseases such as cancer, cardiovascular disease, metabolic disease, and cirrhosis [1,2,3,4,5]. The main active compounds in coffee that play these roles include caffeine, chlorogenic acid, cafestol and kahweol [6,7].

Cafestol and kahweol are natural diterpenes extracted from coffee beans, which mainly present as fatty esters in unfiltered coffee like Turkish coffee [8]. Unfiltered coffee contains 3–6 mg of these diterpenes per cup [9]. The only difference of their structures is that kahweol has an extra double bond [10] (Figure 1). In vivo, About 70% of the consumption of cafestol and kahweol can be absorbed in small intestine [11]. In theory, glucuronidation and sulphation are the major pathways of xenobiotic biotransformation in mammalian species, which occurs largely in the liver, to produce water-soluble products that can be excreted by urine [11]. However, only about 1% or less of the consumption was detected to be excreted as conjugate of glucuronic or sulphuric acid in urine in previous studies [11,12]. It indicates that the major part of the absorbed cafestol and kahweol must be subject to more extensive metabolism, not just by glucuronidation or sulphation. There is evidence that cafestol and/or its metabolites largely accumulate in liver and gastrointestinal tract through the enterohepatic cycle [13]; triggering a variety of biological effects, including secondary changes in liver metabolism.

Early studies confirmed that coffee diterpenes (especially cafestol), effectively increase human plasma triacylglycerol and low-density lipoprotein (LDL), may be a potential risk of inducing some cardiovascular diseases [7,14]. However, from a more comprehensive perspective, cafestol and kahweol show a remarkable two-faced effect. In addition to the deleterious effects on serum lipid levels and liver enzymes in some cases, extensive studies have demonstrated that cafestol and kahweol exhibit a wide variety of pharmacological activities, including anti-inflammatory, anti-angiogenic and anti-tumorigenic properties. With multiple bioactivities of cafestol and kahweol reported, the development of potential multi-target drugs with diterpenes should be encouraged. Therefore, it is of great significance to summarize the pharmacological and biological studies on the two compounds, review the mechanisms behind the effects, and get a comprehensive picture of their miscellaneous functions.

## 2. Biological and Pharmacological Activities of Cafestol and Kahweol

### 2.1. Raising Serum Lipid

Since Thelle et al. first found a positive correlation between coffee consumption and serum concentration of total cholesterol and triglycerides in 1983, many studies have supported their findings over the years [15]. It is confirmed that long-term consumption of unfiltered coffee will effectively cause the increase of plasma triacylglycerol and low-density lipoprotein (LDL) cholesterol in humans [14]. The two diterpenoids, cafestol and kahweol, are mainly responsible for this effect. Plasma lipids, especially LDL cholesterol, are important factors in promoting the occurrence of cardiovascular diseases.

In humans, cafestol appears to be more effective on elevating serum lipid than kahweol. Meta-analyses of observational researches showed that after daily treatment with 10 mg of cafestol for 4 weeks, serum cholesterol increased by 0.13 mmol∙L^−1^, while kahweol raised that by 0.02 mmol L^−1^. LDL cholesterol accounts for about 80% of elevated serum cholesterol. Similarly, the same dose of cafestol is more effective than kahweol to increase serum triglyceride levels. In the test of volunteers, cafestol increased serum triglycerides by 86%, but the combination of cafestol and kahweol increased the response only 7% higher than that of cafestol alone. With the long-term intake of coffee diterpenes, most of the rising serum triglycerides will subside [16,17].

The metabolism of LDL is primarily mediated by the LDL receptor pathway. LDL receptors located on the cell membranes are involved in the endocytic process of lipoproteins containing apolipoprotein B and apolipoprotein E, thereby inducing LDL cholesterol to be eliminated from the bloodstream. Some studies indicate that the LDL receptor may be the active site for cafestol and kahweol on cholesterol-raising effect. In vitro studies suggested that cafestol elevated plasma LDL-cholesterol level at least by suppressing the LDL receptor activity, including decreasing the binding, uptake and degradation of LDL [10,18]. Cafestol did not affect the level of LDL receptor genes but showed a post-transcriptional regulation on LDL receptor [10,18]. Later it was confirmed in vivo by studying on LDL-receptor knockout transgenic mice. These studies showed that cafestol suppressed major regulatory enzymes (such as sterol 27-hydroxylase and oxysterol 7alpha-hydroxylase) in the bile acid synthesis process by activating the nuclear receptors FXR and PXR, leading the decrease of major enzymes mRNA levels [19,20,21]. It may provide a logical explanation for cafestol in raising cholesterol of humans. In addition, elevated plasma lipid transfer proteins levels may be associated with high serum LDL cholesterol concentrations [22]. Cafestol is the predominant factor affecting cholesterolester transfer protein (CETP) and phospholipid transfer protein (PLTP), because the addition of kahweol has little effect on the activity of serum CETP and PLTP. The mixture of cafestol and kahweol can significantly reduce the activity level of lecithin cholesterol acyltransferase and does not appear when acting alone [23]. Many studies have demonstrated that kahweol shows little effect on serum lipid when it works alone [21,24].

### 2.2. Anti-inflammation

Inflammation is a defensive response of bodies to stimulation, which is primarily mediated by activated inflammatory corpuscle. Macrophages produce pro-inflammatory cytokines (such as IL-1 and TNF-α) or inflammatory mediators (NO and PGE2) in the process of pathogen invasion, and plays an indispensable role in boosting initial defense. Although this immune response is a natural defensive mechanism, excessive activation of inflammation is clearly associated with numerous immune diseases such as allergic reactions, autoimmune diseases, infectious diseases, cardiovascular disease, atherosclerosis and cancer.

Lipopolysaccharide (LPS)-induced macrophages activate a variety of inflammatory signaling pathways, which can be widely used to evaluate anti-inflammatory effects. NO and PGE2 are important inflammatory modulators produced by LPS-induced macrophages, and the induction of iNOS and COX-2 significantly increase the level of their synthesis. Kim et al. [25] have found that cafestol and kahweol can significantly inhibit PGE2 and NO synthesis in LPS-activated macrophages with a dose-dependent manner. Cytotoxicity assays on the RAW 264.7 macrophages with cafestol and kahweol showed that these compounds had no adverse effects on cell viability (>90% cell viability). RT-PCR and WB demonstrated that cafestol and kahweol reduced the mRNA levels of *COX-2* and *iNOS*, and decreased the expression of COX-2 and iNOS protein, therefore inhibiting the synthesis of PGE2 and NO. Using an acute air pouch inflammation model induced by carrageenan in a rat, they confirmed that kahweol also had a significant anti-inflammation effect in vivo [26].

Further experiments showed that preventing the activation of NF-κB is the primary mechanism. NF-κB is a heterodimeric protein that generally consists of two functional subunits (P65 and P50) and binds to its natural inhibitor IKB-a/b, which prevents NF-κB entering nucleus. The activation of the inhibitor κB kinase (IKK) is induced by a cascade of events, which phosphorylates IkB, leading to its degradation by the ubiquitin proteasome pathway and finally resulting in the translocation of NF-κB to the nucleus and activation. According to studies [25,27], the coffee diterpenes can inhibit the activation of IKK in LPS-induced macrophages in a dose-dependent manner (within the concentration range of 0.5–10 μM). The IKK complex contains multiple binding sites, which can be regulated by NIK, Akt and MAPK [28]. It is considered that the compounds may inhibited one or more of the IKK upstream kinases to block the activation of IKK rather than reduce IKK expression directly [25,29]. In addition, kahweol has a stronger effect than cafestol on inhibiting PGE2 production and COX-2 expression. Only low doses of kahweol (0.5 μM) can significantly reduce COX-2 levels. Previous studies have indicated that the presence of an extra conjugated double bond in furan ring of kahweol increases its sensibility to electrophilic attack and oxidizing reaction [30]. Therefore, kahweol might be more effective in antioxidant activity. However, the relationship between the different structures of cafestol and kahweol and their efficacy are still unclear, so further specific studies are necessary.

Shen et al. [31] proposed a new point of view that cafestol can effectively block the AP-1 pathway to reduce PEG2 production. AP-1 is a heterodimer composed of c-Fos and c-Jun. In theory, the activation of AP-1 is mainly regulated by phosphorylation of the MAPK signaling pathway. In their experiment, the translocation of c-jun and phosphorylated c-jun into nuclear were effectively blocked after treatment with 25 μM and 100 μM of cafestol. Unexpectedly, this effect of cafestol is not caused by inhibiting the phosphorylation of ERK, JNK and p38, but directly reduces the activity of ERK2 enzyme while sparing ERK1, P38 and JNK. Therefore, they concluded that cafestol blocked AP-1 translocation by inhibiting ERK2, so it could resist PGE2 production. Later they discovered that the pharmacological targets of kahweol were different [32]. Treatment with 100 μM of kahweol did reduce the translocation of p65 to the nucleus but did not affect AP-1(c-jun and c-fos) translocation. Kahweol down-regulates phosphorylation of signal transducers and activators of transcription 1 (STAT1) without altering its total level. This may be caused by inhibiting the phosphorylation of janus kinase 2 (JAK2), which is an upstream kinase of STAT1 [33]. However, the inhibition of JAK2 phosphorylation by kahweol was not stronger than that of JAK2 inhibitor AG490, so there may be other pathways to block phosphorylation of STAT1.

Endothelial cells produce various inflammatory mediators during chronic inflammation, and exacerbate endothelial dysfunction, which is critical in cardiovascular diseases. Mechanical stretching, particularly cyclic strain in vascular endothelial cells, has been confirmed that it can induce the production of ROS, and lead to the increase of cytokines and cell adhesion molecules, including IL-8, MCP-1, and ICAM-1 [34,35]. Recently, Hao et al. [36] first discovered that cafestol could inhibit the secretion of inflammatory molecules induced by cyclic strain in human umbilical vein endothelial cells (HUVECs), possibly through activating HO-1 and Sirt1. In their study, cafestol attenuates MAPK phosphorylation induced by cyclic strain through inhibiting ROS production. In addition to direct antioxidant activity, cafestol may also activate the Nrf2/HO-1 pathway, increase the expression of HO-1, and eliminate excessive free radicals produced by cyclic strain. Furthermore, cafestol upregulates the expression of sirt1 in HUVECs after treatment with cyclic strains. This is another important pathway for cafestol to inhibit cyclic-strain-induced ICAM-1, IL-8, and MCP-1 protein secretion.

### 2.3. Anti- carcinogenesis

Carcinogenesis is a complex and progressive process, which are typically classified into three stages: initiation, promotion, and progression [37]. Extensive studies have shown that cafestol and kahweol have an inhibitory effect at multiple stages of cancer development, including preventing the initiation of tumorigenesis, inhibiting tumor cell proliferation, and tumor metastasis.

#### 2.3.1. Inhibition for Early Mutagenic Event

Early studies have confirmed that coffee diterpenes can prevent cancer by modulating a variety of enzymes involved in carcinogenesis and detoxification. It is known that the electrophilic metabolites or oxidative metabolites produced by activated carcinogens will result in permanent modification of DNA and finally lead to the formation of tumor. And the activation of phase I activating enzymes is a precondition for the activation of some carcinogens. Hepatic cytochrome P450s (CYP450s) plays an important role in the activation of various carcinogens. For example, heterocyclic amines, especially PhIP, can be activated by CYP1A2 and SULT [38]. Similarly, the activation of nitrosamines and other alkylating carcinogens are typically relevant with CYP2E1 or CYP2B1/CYP2B2 [39]. Both in vivo and in vitro studies have demonstrated that the combination of cafestol and kahweol can down-regulate the expression of aforesaid CYP450s or directly inhibit their activity [40,41,42,43,44]. Besides, members of the Phase II detoxification enzyme act by inhibiting the formation of an electrophilic or oxidant intermediate and stimulating its detoxification, thereby reducing DNA damage and preventing tumor initiation [45]. The regulation of these two types of enzymes is a potential way for anti-tumor treatment.

Previous studies indicate that the combination of cafestol and kahweol promotes glutathione S-transferase activation and increases several other phase II xenobiotic metabolizing enzymes such as uridine 5′-diphospho-glucuronosyltransferase and quinine oxidoreductase1 activities [42,43,44,46,47,48,49,50]. Extensive studies have confirmed that cafestol and kahweol can intervene Keap1/Nrf2/ARE signaling pathways, and induce the expression of phase II detoxifying enzymes, antioxidant proteins and Nrf2, thus blocking several carcinogenic stages mentioned above [44]. This detoxification effect of cafestol and kahweol in liver have been reviewed in detail by Tao et al., indicating that it has the potential to prevent the occurrence of various carcinogenesis, liver cancer in particular [45,49,51].

#### 2.3.2. Inducing Apoptosis

Apoptosis is a complex process involving the activation, expression, and regulation of a series of genes such as the Bcl-2 protein family and cell cycle regulatory proteins. Bcl-2 family proteins are both activators and inhibitors, playing a pivotal role in the process of apoptosis. Another marker of the apoptotic pathway is the cleavage of caspase-3, which is the most important terminal cleavage enzyme in apoptosis [52].

In recent years, a large number of studies have shown that cafestol and kahweol can significantly inhibit tumor cell activity, regulate cyclins and apoptosis-related proteins through multiple targets, thereby inhibiting tumor cell proliferation and inducing apoptosis [53,54,55,56].

Choi et al. [56] observed that cafestol inhibited the proliferation and induced apoptosis in Caki cells, of which the effect was positively correlated with concentration of cafestol (10–40 μM). The mechanism behind is down-regulating the expression of anti-apoptotic proteins (Bcl-2, Bcl-xL, mcl1, cFLIP) and the level of mitochondrial membrane potential, and released cytochrome c. Cafestol-induced apoptosis also shows a dependence on the activation of caspases, a key executioner of apoptosis. In addition, their study also demonstrated that cafestol inhibited PI3K/Akt pathway, and exhibited significant synergy with PI3K inhibitor LY29004 in Caki cells. Similarly, Woo et al. [54] reported that cafestol significantly enhanced the therapy sensitivity of ABT-737 in several common cancer cell lines and promoted their apoptosis by regulating Bcl-2 family proteins. These findings suggest that cafestol is a potential treatment for cancers such as kidney cancer. Specificity protein 1(Sp1) is an important transcription factor for regulating tumor growth, and also the binding site of many promoters of apoptosis-related genes [57]. Lee et al. [55] considered that cafestol (30–90 μM) and kahweol (20–60 μM) suppressed Sp1 protein expression to down-regulate apoptotic proteins through a study on human malignant pleural mesothelioma. Recently, Lima et al. [52] have found that cafestol (80 μM) can reduce the production of ROS in the leukemia cell line HL60. Moreover, the differentiation markers CD11b and CD15 increase with cafestol treatment at low doses (10 μM). This may help induce apoptosis and promote leukemia cell differentiation. More interestingly, cafestol inhibits the clonogenic potential of HL60 cells but does not affect normal murine CFU-GMC. Woo et al. [54] have also demonstrated that cafestol cannot impair the viabilities of normal human fibroblast cells, indicating a more selective impairment of tumor cells.

Kahweol, as a derivative of cafestol, also shows similar anti-cancer activity. Studies on various tumor cells have confirmed that kahweol-induced apoptosis is related with caspase 3 activation, cytochrome c releasing from mitochondria to cytoplasm and down-regulating anti-apoptotic proteins and cyclins [58,59,60,61,62]. Kahweol plays anti-cancer effects by regulating Sp1, Akt, ERK/JNK pathways with multiple targets [58,61,62,63,64,65]. In addition to the above mechanism, Kim et al. [65] also reported that kahweol inhibited proliferation and promoted apoptosis in A549 cells in a positively correlation with treated time (24–48 h) and dose (10–40 μM). For the first time, they reported that kahweol down-regulated the STAT3 signaling pathway by inhibiting its constitutive phosphorylation and activation except protein levels. In addition, activating transcription factor 3 (ATF3) is a reactive protein produced under cellular stress conditions, which can enhance the activation of p53 [66], suppress Ras-mediated tumorigenesis [67], and down-regulate cyclin D1 and MMP-2 expression [68]. Park et al. [69] found that kahweol induced apoptosis by upregulating ATF3 in human colorectal cancer cells, which might depend on the activation of ERK1/2 or GSK3β. Basic transcription factor 3(BTF3) is involved in the apoptotic process of various cells and exhibits over-expression in malignant tumors. It is reported that kahweol inhibits BTF3 expression through the ERK-mediated signaling pathway to induce apoptosis of non-small cell lung cancer cells [70]. Kahweol, as same as cafestol, particularly inhibits the tumor cells activity, but has no effect on normal cells [60,71].

In addition to the independent anticancer activity, the two coffee diterpenes also work synergistically with several anticancer drugs. According to previous studies, cafestol effectively overcomes the resistance to ABT-737 of human glioma U251MG cells and human breast carcinoma MDA-MB231 cells [54]. The combination of cafestol and Ara-C reduced the viability of HL60 cells more efficiently compared with the two drugs used alone [52]. Kahweol and sorafenib showed a synergistic effects in inducing apoptosis on Caki cells [71]. These results suggest that cafestol and kahweol have a great potential in the combined therapy with anticancer drugs.

#### 2.3.3. Anti-Angiogenesis

In 1971, Folkman first proposed the idea that tumor growth was dependent on angiogenesis [72]. Angiogenesis plays an important role in tumor invasion and metastasis. Studies have shown that the degree of tumor angiogenesis is significant for histopathological grading and prognosis [73]. Therefore, Anti-angiogenesis is one of the hot spots for cancer treatment.

Angiogenesis is a complex multi-step process. Normal endothelial cells (EC) are at quiescent. Under pathological conditions such as tumors, quiescent ECs are activated by some pro-angiogenic signals, which lead to EC proliferation, proteolysis of extracellular matrix (ECM), endothelial cell migration, anastomosis, and ultimate recruitment of supporting cells. Targeting any of these key steps can be an effective way to inhibit angiogenesis. Cardenas et al. [74] demonstrated the anti-angiogenic effect of kahweol both in vitro and in vivo. In chicken chorioallantoic membrane assay, they observed that 50 nM of kahweol inhibited angiogenesis in each treated egg. They also found that after being treated with 25 μM kahweol, 75% of larvae showed inhibited angiogenesis in the transgenic zebrafish model. In vitro, only 5 μM kahweol can clearly inhibit endothelial cell sprouting in the mouse aortic ring assay. In cell experiments, kahweol inhibits the proliferation, tubule formation and migration of HUVECs in a dose-dependent manner (25–75 μM). Besides, kahweol is able to inhibit the expression of MMP2 and uPA, which are two relevant enzymes involved in ECM remodeling [75], thus inhibiting the invasion ability of ECs.

Vascular endothelial growth factor (VEGF) is the most important growth factor during angiogenesis. There is a consensus now that VEGF receptor-2(VEGFR-2) is the primary mediator of pro-angiogenic effect of VEGF [76]. Focal adhesion kinase (FAK) is a signaling molecule which is highly correlated with tumor invasion and metastasis and has been confirmed to regulate the proliferation and migration of ECs by the VEGFR-2 pathway [77]. Their downstream mediators, including Akt, Erk, and NO, are feasible targets for anti-angiogenic therapy. A vitro study showed that 20–80 μM of cafestol inhibited HUVECs proliferation in a dose-dependent manner. And the migration and tube formation of HUVECs were clearly inhibited after treatment with 5–20 μM of cafestol. Further experiments demonstrated that 20 μM of cafestol significantly inhibited FAK and Akt phosphorylation and NO production, and partly inhibited VEGFR2 phosphorylation in the HUVECs. Therefore, they concluded that cafestol could exhibit anti-angiogenic properties mainly by regulating the VEGF downstream such as the phosphorylation of Akt and FAK and NO production [78]. Moeenfard et al. [79] assessed the anti-angiogenic activity of cafestol palmitate (CP) and kahweol palmitate (KP) on human microvascular endothelial cells (HMVECs). Their study showed that both CP and KP at 50 μM inhibited the proliferation and migration of HMVECs through down-regulating the expression of VEGFR2 and Akt. In most assays, KP exhibited stronger anti-angiogenic properties than CP. This phenomenon might also due to the extra double bond in kahweol mentioned earlier.

Urotensin II is known as the strongest vasoconstrictor. It has been confirmed that urotensin II shows a strong pro-angiogenic response and promote the expression of other pro-angiogenic factors such as VEGF and interleukin-8 (IL-8) [80,81]. Recently, Tsai et al. [82] proposed a new interpretation of the regulation mechanism of cafestol on vascular endothelial cells. Their study demonstrates that pretreatment with 3–10 μM of cafestol (12 h) can significantly inhibit the IL-8 secretion in urotensin II-induced HUVECs. They also observed that the mRNA and protein levels of HO-1 are positively correlated with cafestol treated time (3–24 h) and dose (3–10 μM). They concluded that cafestol might induce HO-1 to decrease IL-8 expression induced by urotensin II, thereby suppressing ECs proliferation.

### 2.4. Potential Anti-Diabetic Capabilities

Type-2-diabetes (T2D) is described as a relative insulin deficiency, with obesity one of the risk factors. Previous studies have shown that moderate coffee consumption may prevent the development of T2D and confirmed that caffeic acid and chlorogenic acid have the potential to decrease the risk of T2D.

Mellbye et al. [83] first discovered that 10-8 M of cafestol acutely stimulated clonal rat insulinoma cell line INS-1E to secrete insulin and also increased glucose uptake in muscle cells. The motivating effect of cafestol on glucose uptake is similar to that of rosiglitazone, but the underlying pharmacological mechanism needs to be further clarified. It was surprising that oxokahweol did not show parallel effects on stimulating insulin secretion in this study, for which the reason remains unknown. Subsequently, they used KKAy mice (a T2D murine model) for in vivo studies [84]. After cafestol intervention for 10 weeks, both low-dose (0.4 mg/day) and high-dose (1.2 mg/day) groups significantly reduced fasting blood glucose, fasting glucagon, promoted insulin secretion and increased its sensitivity in KKAY mice.

In a recent study, Baek et al. [85] observed that kahweol (25 μg/mL) reduced lipid accumulation in 3T3-L1 cells. This result was confirmed to be achieved by inhibiting the expression of adipogenesis and lipid accumulation-related genes including PPARγ, C/EBPα, FABP4, and FASN. Kim et al. also got a similar conclusion [86]. Further tests indicate that kahweol can activate Adenosine 5‘-monophosphate-activated protein kinase (AMPK) to inhibits lipid accumulation and stimulate glucose uptake. AMPK is a central modulator of metabolism of lipid and glucose. Therefore, activating AMPK is a promising therapy for diabetes and other metabolic diseases. It is well known that metformin, a predominant drug for T2D, is a typical AMPK activator [87].

In general, previous studies have indicated the potential anti-diabetic effects of cafestol and kahweol, but their different efficacy and mechanism of action remain unclear. Therefore, further researches and clinical trials are needed to confirm whether the two coffee diterpenes can be used to prevent or treat T2D in humans.

### 2.5. Anti-osteoclastogenesis

Osteoclasts (OCs) and osteoblasts (OBs) play an important role in bone formation and remodeling [88]. The imbalance between OCs and OBs can cause degenerative bone diseases such as osteoporosis, bone metastasis of cancer, arthritis, osteopetrosis, etc. [89].

Fumimoto et al. [90] found that kahweol can inhibit the differentiation of OCs in a dose-dependent manner. The mechanism behind it should be dependent on its anti-inflammatory and antioxidant effects. Kahweol completely blocked the phosphorylation of ERK and partly inhibit phosphorylation of Akt stimulated by receptor activator of nuclear factor kappa-B ligand (RANKL). Besides, kahweol down-regulates the transcription of OCs makers (Src and cathepsin K) via impairing NFATc1 expression. They also observed that kahweol promoted HO-1 expression, a phase II antioxidant enzyme. In later research, they confirmed that cafestol also showed similar ability to inhibit OCs differentiation and bone resorbing activity [91]. Same as kahweol, cafestol suppress NFATc1, Src and cathepsin K expression. It is worth noting that kahweol has a stronger inhibitory effect on osteoclastogenesis than cafestol. This may be because the extra double bond of kahweol makes it more prone to epoxidation, thus promoting the expression of phase II antioxidant enzymes through Nrf2/ARE pathway. In addition to inhibiting OCs differentiation, cafestol promotes OBs differentiation. However, this dual effect on bone cell proliferation has not been mentioned on kahweol. The inhibitory effect on osteoclastogenesis indicate that cafestol and kahweol have potential to develop new drugs for degenerative bone diseases treatment.

## 3. Conclusions

Although cafestol and kahweol have certain adverse effects on raising serum lipids, more attention should be paid to its extensive anti-inflammatory, anti-cancer, and other potential pharmacological activities (Figure 2). Cafestol and kahweol can regulate a variety of inflammatory mediators to reduce inflammation. In addition, the two coffee diterpenes can prevent cancer from occurring by blocking the activation of carcinogens and improving liver detoxification function. They can also inhibit tumor cell proliferation and angiogenesis, and provide a new approach for cancer prevention and treatment. At present, some mechanism of bioactivities and the difference of cafestol and kahweol are still unclear, as seen in Table 1. Therefore, further studies are needed to clarify the pharmacological effects and mechanisms of action, and solve their safety issues, thereby making the compounds potential multi-target complementary medicines and functional food.

## Figures and Tables

**Figure 1 ijms-20-04238-f001:**
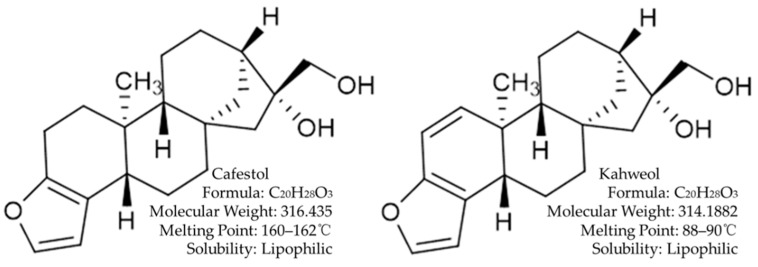
Structure and chemical characteristics of cafestol and kaweol. They are natural diterpenes extracted from coffee beans.

**Figure 2 ijms-20-04238-f002:**
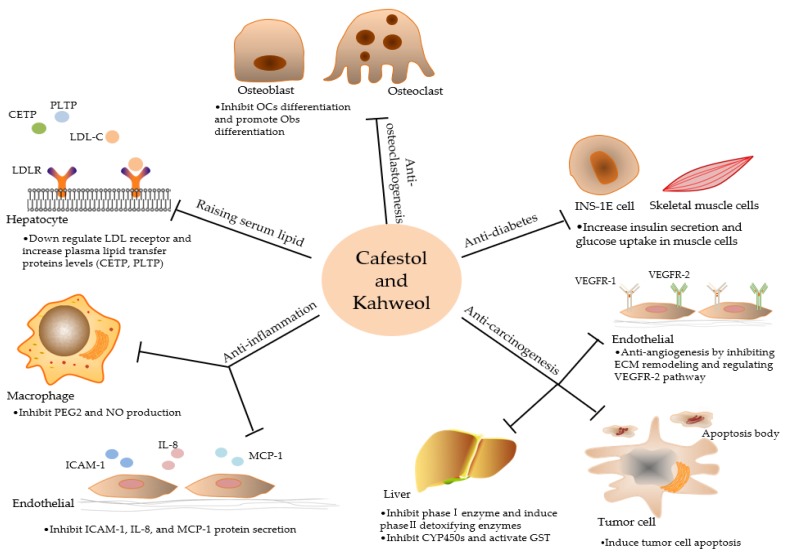
Diagram showing bioactivities and targets of cafestol and kahweol. Cafestol and kahweol raise human serum lipid level, and show extensive anti-inflammatory, anti-cancer and potential anti-diabetic activities.

**Table 1 ijms-20-04238-t001:** Effects of cafestol and kahweol.

Activities	Mechanism	Comparison of efficiency
Raising serum lipid	•Down regulate LDL receptor and increase plasma lipid transfer proteins levels (CETP, PLTP)	Cafestol is far stronger than kahweol
Anti-inflammation	•Inhibit the expression of iNOS and COX-2 and the secretion of pro-inflammatory cytokines	Kahweol might be more effective in antioxidant activity
	•Inhibit phase I enzyme and induce phase II detoxifying enzymes: Nrf2/ARE	
Anti-carcinogenesis	•Induce apoptosis by regulating Bcl-2 family proteins and cyclins •Anti-angiogenesis by inhibiting ECM remodeling and regulating VEGFR-2 pathway	Kahweol exhibited stronger anti-angiogenic properties than cafestol in some studies
Anti-diabetes	•Increase insulin secretion and glucose uptake in muscle cells •Inhibit adipogenesis	Not mentioned
Anti-osteoclastogenesis	•Inhibit differentiation and bone resorbing activity of OCs •Promote OBs differentiation	Kahweol stronger in inhibiting osteoclastogenesis than cafestol

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
