# Peer review of "Cafestol and Kahweol: A Review on Their Bioactivities and Pharmacological Properties"

_ijms, 2019, doi:10.3390/ijms20174238_

Round 1
Reviewer 1 Report
The authors described thoroughly the Cafestol and Kahweol biological properties. A few concerns should be addressed before publishing this manuscript.
CONCERNS:
1/INTRODUCTION ” …then extensively metabolized to the whole body by liver..” The authors need to clarify that point. It is not clear , and additionally references about metabolism of the two compounds should be added in the text.
2/ ANTI-CARCINOGENESIS “Early studies have confirmed that coffee diterpenes can prevent cancer by modulating a variety of enzymes involved in carcinogenesis and detoxification. It is known that the electrophilic metabolites or oxidative metabolites produced by activated carcinogens will result in permanent modification of DNA and finally lead to the formation of tumor. And the activation of phase I activating enzymes is a precondition for the activation of some carcinogens. Hepatic cytochrome P450s (CYP450s) plays an important role in the activation of various carcinogens. …." References should be added in the text. Beside, it is too general. Not every isoenzyme of P450 can be connected to activation of carcinogens.
3/ ANTI-DIABETIC CAPABILITIES " Mellbye et al[72]. first discovered that 10-8 M dose of cafestol,…" It is a conc. Unit, not a dose.
4/ The authors could add information about the effect of two title compounds on osteoclastogenesis.
5/ The references section should be corrected. For example reference no 27 or 42.
Author Response
Thank you very much for your great suggestions.
1/INTRODUCTION” …then extensively metabolized to the whole body by liver..” The authors need to clarify that point. It is not clear, and additionally references about metabolism of the two compounds should be added in the text.
Response 1: Thank you very much. This part had been redrafted in our manuscript according to your suggestions.
“In vivo, About 70% of the consumption of cafestol and kahweol can be absorbed in small intestine[11], In theory, glucuronidation and sulphation are the major pathways of xenobiotic biotransformation in mammalian species, which occurs largely in the liver, to produce water-soluble products that can be excreted by urine[11]. However, only about 1% or less of the consumption was detected to be excreted as conjugate of glucuronic or sulphuric acid in urine in previous studies [11-12]. It indicates that the major part of the absorbed cafestol and kahweol must be subject to more extensive metabolism, not just by glucuronidation or sulphation. There is evidence that cafestol and /or its metabolites largely accumulate in liver and gastrointestinal tract through enterohepatic cycle[13], triggering a variety of biological effects,including secondary changes in liver metabolism.”
2/ ANTI-CARCINOGENESIS “Early studies have confirmed that coffee diterpenes can prevent cancer by modulating a variety of enzymes involved in carcinogenesis and detoxification. It is known that the electrophilic metabolites or oxidative metabolites produced by activated carcinogens will result in permanent modification of DNA and finally lead to the formation of tumor. And the activation of phase I activating enzymes is a precondition for the activation of some carcinogens. Hepatic cytochrome P450s (CYP450s) plays an important role in the activation of various carcinogens. …." References should be added in the text. Beside, it is too general. Not every isoenzyme of P450 can be connected to activation of carcinogens.
Response 2: Thank you very much. This part had been redrafted in our manuscript according to your suggestions.
“Early studies have confirmed that coffee diterpenes can prevent cancer by modulating a variety of enzymes involved in carcinogenesis and detoxification… For example, heterocyclic amines, especially PhIP, can be activated by CYP1A2 and SULT [38]. Similarly, the activation of nitrosamines and other alkylating carcinogens are typically relevant with CYP2E1 or CYP2B1/CYP2B2[39]. Both in vivo and in vitro studies have demonstrated that the combination of cafestol and kahweol can down-regulate the expression of aforesaid CYP450s or directly inhibit their activity [40-44].”
3/ ANTI-DIABETIC CAPABILITIES" Mellbye et al[72]. first discovered that 10-8 M dose of cafestol,…" It is a conc. Unit, not a dose.
Response 3: Thank you very much. This part had been corrected in our manuscript according to your suggestions.
4/ The authors could add information about the effect of two title compounds on osteoclastogenesis.
Response 4: Thank you very much. This part had been added in our manuscript according to your suggestions.
“2.5. Anti-osteoclastogenesis”
5/ The references section should be corrected. For example reference no 27 or 42.
Response 5: Thank you very much. The references were corrected.
Reviewer 2 Report
Introduction is definitely too brief. As coffee is main food source of diterpens, it should be considered in introduction. Please briefly mention the main health effects of coffee toward human health confirmed by meta-analysis of prospective cohort studies, as they can be (at least partially) mediated by diterpens.
Introduction please add the information about effect of diterpens on blood lipids (PMID: 25046596)
Author Response
Thank you very much for your great suggestions.
Introduction is definitely too brief. As coffee is main food source of diterpens, it should be considered in introduction. Please briefly mention the main health effects of coffee toward human health confirmed by meta-analysis of prospective cohort studies, as they can be (at least partially) mediated by diterpens.
Response 1: Thank you very much. This part had been redrafted in our manuscript according to your suggestions.
“Coffee is one of the most wildly consumed beverages in the world. Extensively experiments and meta-analysis of studies show that coffee consumption is associated with the risk of some human diseases [1]. Statistics show that moderate consumption of coffee (3 to 5 cups/day) reduces the risk of cardiovascular disease [2]. There is growing evidence that coffee can prevent some chronic diseases such as cancer, cardiovascular disease, metabolic disease and cirrhosis [1-5]. The main active compounds in coffee that play these roles include caffeine, chlorogenic acid, cafestol and kahweol [6-7].”
Introduction please add the information about effect of diterpens on blood lipids (PMID: 25046596)
Response 2: Thank you very much. This part had been redrafted in our manuscript according to your suggestions.
“Early studies confirmed that coffee diterpenes (especially cafestol) effectively increase human plasma triacylglycerol and low-density lipoprotein (LDL), may be a potential risk of inducing some cardiovascular diseases [7, 14].”
Round 2
Reviewer 2 Report
Thank you for revising manuscript